# Pharmacological and Nutritional Modulation of Metabolome and Metagenome in Cardiometabolic Disorders

**DOI:** 10.3390/biom13091340

**Published:** 2023-09-02

**Authors:** Anna Maria Witkowska, Joe-Elie Salem

**Affiliations:** 1Department of Food Biotechnology, Faculty of Health Sciences, Medical University of Bialystok, Szpitalna 37, 15-295 Białystok, Poland; 2Department of Pharmacology, Pitié-Salpêtrière Hospital, Institut National de la Santé et de la Recherche Médicale (INSERM), Clinical Investigation Center (CIC-1901), Assistance Publique-Hôpitaux de Paris (AP-HP), Sorbonne Université, 75013 Paris, France; joe-elie.salem@aphp.fr

**Keywords:** cardiometabolic, drugs, food, supplement, diet, nutrition, metabolome, metagenome, microbiota

## Abstract

Cardiometabolic disorders are major causes of morbidity and mortality worldwide. A growing body of research indicates that the gut microbiota, whether it interacts favorably or not, plays an important role in host metabolism. Elucidating metabolic pathways may be crucial in preventing and treating cardiometabolic diseases, and omics methods are key to studying the interaction between the fecal microbiota and host metabolism. This review summarizes available studies that combine metabolomic and metagenomic approaches to describe the effects of drugs, diet, nutrients, and specific foods on cardiometabolic health and to identify potential targets for future research.

## 1. Introduction

Cardiometabolic disease (CMD) is associated with multimorbidity and increased mortality worldwide [1,2,3,4]. The prevalence of CMD varies between countries and continents [2]. In Canada, the overall prevalence of CMD is 3.5% [1]. In China, the multimorbidity of CMD increased from 2.41% to 5.94% between 2010 and 2016 [3]. The growing population with metabolic disorders can become a challenge for the national healthcare system.

In the literature, CMD has been described as a cluster of interrelated risk factors such as abdominal obesity, hypertension, elevated fasting plasma glucose, and dyslipidemia [5,6], which leads to cardiovascular disease, stroke, and type 2 diabetes mellitus (T2DM) [7]. Unhealthy diets, inactivity, and smoking are among the major risk factors for CMD [6,7]. The individual symptoms that make up cardiometabolic disease require an integrated approach and must be pharmacologically treated and prevented by increasing activity and improving diet composition [8]. 

An inadequate diet combined with low physical activity promotes obesity, hypertension, type 2 diabetes, insulin resistance, and dyslipidemia [9,10]. Obesity, especially visceral obesity, leads to body fat accumulation, lipotoxicity, inflammatory and oxidative process initiation, and T2DM [11]. Among dietary modifications to reduce or prevent cardiovascular disease, caloric restriction in obesity is essential to achieving and maintaining weight loss [12]. It normalizes inflammatory markers, lowers blood pressure, and modulates insulin response and glucose levels [12,13]. An essential element of nutrition in preventing cardiovascular disease is the qualitative composition of the diet, which involves increasing the intake of vegetables, fruits, legumes, nuts, and seeds [12,13]. Plant products contain many components with antioxidant activity that modulate the composition of the intestinal microbiota, including polyphenols and dietary fiber. Currently, more and more new plant raw materials containing biologically active substances are being sought, which could be ingredients in diets, medicines, or supplements for people with metabolic disorders and cardiovascular diseases [14].

Polypharmacy is common in CMD [15]. Among the essential medications for treating individual symptoms are drugs for diabetes, high blood pressure, appetite control, and cholesterol-lowering [15]. In treating obesity and NAFLD, drugs that control appetite and inhibit steatohepatitis, such as liraglutide, have a promising role. In the monotherapy of type 2 diabetes, metformin is the treatment of choice in the early stages. Depending on the symptoms of cardiovascular disease, patients require high blood pressure medications, anticoagulants, aspirin, and, for gastroenterological side effects, proton pump inhibitors (PPIs) [15]. Frequently, antibiotics are administered because of the increased prevalence of infections [16].

Some of the effects of nutrition and pharmacotherapy may go beyond the impact of food and drug chemicals alone as a result of the activity of the gut microbiota, which transforms chemicals in the gut and, through the resulting compounds, exerts systemic effects and modulates the effects of treatment. The gut is home to trillions of microorganisms such as bacteria, archaeons, fungi, and viruses, of which bacteria are the most numerous and most studied group. The gastrointestinal tract is populated in 90% by two bacterial phyla: *Firmicutes* and *Bacteroidetes*, followed by *Proteobacteria*, *Actinobacteria*, and *Verrucomicrobia* [17]. In fact, many of the compounds found in the bloodstream result from the activity of the gut microbiota, which converts nutrients, mainly carbohydrates and proteins as sources of carbon and energy, and other food compounds into various chemical derivatives through numerous metabolic pathways [18,19]. A healthy microbiota is necessary to maintain the intestinal barrier. Intestinal bacteria can ferment food and extract nutrients by breaking down macronutrients. The effect of polysaccharide fermentation is the formation of Short Chain Fatty Acids (SCFAs), such as primarily butyric acid, which show beneficial effects on the gastrointestinal tract by providing protection and integrity [18]. By acidifying the environment, SCFAs reduce the ability of pathogens to colonize the digestive tract and increase mucus production [19]. The effects of SCFAs are not limited to the gastrointestinal tract. SCFAs can also modulate glucose metabolism, regulating tissue insulin sensitivity and the body’s anti-inflammatory response [19]. In addition to beneficial effects, microbiota may contribute to the formation of harmful products. Several microbial metabolites exhibit negative effects on the host. Microbiota-derived amino acid metabolites, such as indoxyl sulfate and p-cresol sulfate, promote the development of vascular inflammation [20]. Trimethylamine oxide, produced by the microbiota from choline-rich foods such as red meat, plays an adverse role in the pathology of cardiovascular disease and correlates with increased cardiovascular mortality [21,22,23]. The mechanistic link of this association is platelet hyperresponsiveness and thrombosis [23,24]. Acylcarnitine fecal metabolites, which result from the conjugation of fatty acids with L-carnitine, are positively correlated with obesity [25]. Another aspect of microbiota assessment is to evaluate the abundance of different species of gut microbiota, which provides an opportunity to track the effectiveness of dietary therapy. The literature suggests that a high *Firmicutes* to *Bacteroidetes* ratio may be associated with a higher risk of metabolic syndrome, diabetes, and obesity. Previous studies have indicated that calorie-restricted diets increase the abundance of *Bacteroides* species, which are inversely associated with obesity, high-fat diets, and low dietary fiber [26,27,28]. A whole range of cutting-edge techniques, such as metabolomics, proteomics, nutrigenomics, and metagenomics, which have evolved rapidly over the past two decades, make it possible to learn about these processes (Figure 1). They allow the identification of a significant number of compounds produced by microorganisms and absorbed from the gastrointestinal tract, as well as the study of the metabolic pathways of these compounds in the host body and their potential role in the treatment of various diseases. Due to the multiple metabolic activities of the microorganisms inhabiting the intestines, particularly the production and secretion of hormone-like substances, the microbiota interacts with the body’s endocrine system in tissues and organs. Research indicates that a greater diversity of microbiota promotes better health outcomes [29]. Adverse changes in the intestinal microbiota are referred to as “intestinal dysbiosis,” a state characterized by altered microbial diversity and composition at the α- and β-diversity levels. The α-diversity expressed as richness (number of taxonomic groups) and evenness (distribution of abundances of the groups), is a measure of variation within a microbiome community, while β-diversity measures the similarity or distance between microbiome communities. Dysbiosis disrupts the complex balance of hormones and metabolic processes and leads to pathological processes leading to diabetes mellitus and cardiovascular disease [30]. 

Metagenomics and metabolomics are relatively young scientific disciplines. Metagenomics allows direct cloning, sequencing, and functional analysis of genetic material isolated from the gut microbiome. Unlike bacterial culture studies, the techniques used in metagenetic studies enable the detection of nucleic acids present in the environment, so the isolated genetic material should overlap with the genetic material of microorganisms present in biological samples. Metabolomics allows studying the metabolome of low-molecular-weight compounds in biological samples [31]. It uses advanced technologies such as nuclear magnetic resonance and mass spectrometry to determine amino acids, phosphosugars, SCFA, nucleotides, and their precursors and metabolites. 

An integrated approach to metabolomics and metagenomics can reveal an interplay between microbiota and host metabolism and provides much more material for interpretation than individual omics. Research in recent years has increasingly considered a dual-omics, or even cross-omics, approach to assessing the metabolome and metagenome to predict cardiovascular disease. An example of this combined assessment is the study by Feng et al. [32]. Using metagenomic and metabolomic technologies, the authors found that plasma and urinary N-acetylglucosamine-6-phosphate and urinary mannitol, compounds initially produced by the gut microbiota, may have potential as biomarkers of coronary artery disease. They combined metabolome data with specific species of bacteria inhabiting the gut (*Clostridium* sp. HGF2, *Streptococcus* sp. M143, and *Streptococcus* sp. M334) to find a link between the microbiota and the host metabolome [32].

This qualitative review aimed to identify publications with an integrated approach combining metabolomics and metagenomics to describe the complex impact drug treatment, diet, nutrients, and specific foods can have on cardiometabolic health and identify potential targets for future research. Due to the complex subject matter and the extensive research material described in the studies, the review summarizes the most important findings collected by metabolomics and metagenomics.

## 2. Search Strategy

The search was conducted using PubMed, which covers biomedical literature from Medline, life science journals, and online books. The search was performed in July 2022 and included articles from PubMed’s inception until June 2022. Since the PubMed database did not yet have completed articles in July, articles published in July were not considered for this review. Search terms included: metabolomics, metabolome, metagenomics, metagenome, nutrition, food, supplement, drug, pharmacology, cardiometabolic, and cardiovascular (Figure 2). The articles included in the search strategy were observational studies (cohort and case-control studies), experimental studies (clinical trials), and animal studies. The systematic search was supported by hand searches that included multiple drug-related terms such as metformin, statins, liraglutide, and orlistat. Only those articles where metabolomic and metagenomic techniques were applied in the context of CMD or CMD risk factors were selected for presentation in this review. Full-text articles published in English that were relevant to this review were selected.

## 3. Results

A search by predefined terms yielded 262 hits (Figure 2). Based on these, the abstracts were reviewed, and 116 papers were qualified for further review. After reviewing the full texts of the papers, those that did not address pharmacotherapy in association with hypertension, elevated plasma glucose, dyslipidemia, obesity, non-alcoholic liver fatty disease, CMD, or T2DM, and those that did not mention diet or food in the context of prevention or treatment of CMD, and those that did not present metabolomic and metagenetic studies concurrently, were rejected. Finally, 30 papers from 2013–2022 were included in the review: nine reports concerning pharmacotherapy and 19 dealing with food, supplements, or nutrition. By reviewing the publication dates, it was found that 86% of the collected articles were from 2020–2022. A flowchart of article selection is shown in Figure 3.

### 3.1. Pharmacotherapy

Of the pharmacotherapy-related publications, all nine articles were published between 2020 and 2022, with the majority of them being human studies (seven of the nine articles) and one combining human and animal studies.

These nine publications covered single drugs and groups of medications, such as metformin, statins, orlistat, ezetimibe, antibiotics, or combinations of different drugs and groups of drugs, including statins, the dipeptidyl peptidase-4 inhibitor PKF-275-055, and antibiotics. The publications covered CMD, T2DM, coronary artery disease, ischemic heart disease, NAFLD, and cirrhosis. The articles were compiled and presented in Table 1. 

Statins are commonly used to lower LDL cholesterol levels, reducing cardiovascular risk. They are effective in lowering cholesterol; however, statins cause adverse effects such as impaired metabolic control and an increased risk of T2DM in a particular group of patients. A study by Wilmanski et al. [33], using two independent cohorts, aimed to investigate the potential role of the gut microbiome in modifying patient responses to statin therapy. The authors concluded that HMG (a hydrolyzed substrate of 3-hydroxy-3-menthyl glutarate coenzyme-A reductase) might become a promising marker of statin targeting. Plasma HMG levels reflected the intensity of statin therapy and known genetic markers of variable response to statins. In contrast, heterogeneity in response to statins was consistently associated with variation in the gut microbiome. A gut microbiome enriched in *Bacteroides* and depleted in diversity was associated with more intense responses to statins, both in terms of on-target and adverse effects. 

The effects of drugs on the microbiome and metabolome can be complex, especially in multidrug regimens. The MetaCardis cohort study of more than 2000 Europeans provides information on the interactions between drug groups and gut microbiota [36]. The drug groups in this study included medications most commonly prescribed to CMD patients, such as antidiabetic medications, antihypertensive, antidyslipidemic, antithrombotic, antiarrhythmic, and gout medications, PPIs, and antibiotics. This study evaluated the effects of individual drugs and their combinatorial effects. Univariate statistical analysis was used to separate drug effects from factors related to the disease, vs. gut microbiome, and host characteristics to quantify the impact of individual drugs. Research indicated that much of the naive relationship between drugs and the microbiome, or metabolome, could be attributed to drug intake. Still, not all interactions can be related only to the effects of treatment. Considering the direct impact of drugs, of the 28 cardiometabolic drugs studied, the most potent effect on serum metabolomes was found for antidiabetic drugs, statins, beta-blockers, anticoagulants, and aspirin. This study identified aspirin-associated changes in bacterial species abundance and shifts in the serum lipidome and metabolome associated with improved cardiometabolic health. They included depletion of bacteria associated with inflammatory diseases such as *Ruminococcus gnavus* and *Parvimonas micra*, among others; reduction of plasma inflammatory markers; and a reduction in levels of pyruvate, glutamate, and succinate, which, as previous studies found, are associated with obesity, to an extent comparable to aspirin levels detected in the serum of treated subjects. In addition, γ-butyrobetaine, a proatherogenic intermediate of microbial metabolism, was lower in aspirin takers, revealing a potential complex antiatherosclerotic effect of aspirin beyond its known anti-platelet functions. PPIs had the most associations with gut microbiome features, including a higher prevalence of oral bacteria, probably due to the transfer of bacteria from the mouth to the gut due to reduced gastric acidity. In this study, a synergistic positive effect of the drug combination on disease markers was observed. In T2DM, the most pronounced synergistic effect on microbiome traits was observed with loop diuretics, especially in combination with aspirin, angiotensin-converting enzyme (ACE) inhibitors, and beta-blockers, and on host features with statins. Loop diuretics in combination with aspirin, ACE inhibitors, or beta-blockers were found to more strongly enrich microbiome-related health markers. Statins, taken together with metformin or aspirin, contributed to the microbiome’s richness and abundance of Firmicutes and methanogenic bacteria, which are depleted in T2DM. This is consistent with previous results of a study that showed that statin therapy is associated with a lower incidence of dysbiosis of the gut microbiota [42]. The authors hypothesize that these shifts in the microbiome may mediate some of the synergistic effects of the drugs on the host. Other drugs used for lipid disorders in high-risk patients include ezetimibe and orlistat [39]. Research showed that intestinal orlistat- and ezetimibe-mediated malabsorption of fat and cholesterol from food had limited effects on the overall gut microbial community and their metabolites. During the intervention, the gut microbiota and their SCFA metabolites were relatively stable in overweight and obese Chinese individuals with dyslipidemia. 

Growing evidence shows a link between antibiotic therapy, dysbiosis, metabolic diseases, and cardiovascular risk, independently of diet. A cross-omics study by Kappel et al. [40], conducted in mice and humans, explored the complex interaction between the gut microbiome and host metabolism. They fed mice a Western-type diet (WTD) or a normal chow diet and challenged them with non-absorbable antibiotics (NAA). As they expected, mice fed WTD developed atherosclerotic lesions; however, NAA, independently of the diet, augmented the development of atherosclerosis. A decrease in tryptophan metabolites and other metabolic pathways, mainly related to altered lipid metabolism, was accompanied by a decrease in a few bacterial families, such as those belonging to *Clostridia*, *Ruminococcaceae*, *Lachnospiraceae*, and *Porphyromonadaceae* of *Bacteroidetes.* Additionally, several antibiotics may have a quantitative relationship between the number of antibiotic therapy courses and the microbiome’s deteriorating state, which may reflect the severity of the cardiometabolic disease [36].

Antibiotic use can lead to long-term enrichment of the gut microbiota’s antibiotic resistance genes (ARGs), called the antibiotic resistome. Shuai et al. [35] found a shift between the ARG of healthy subjects, prediabetes, and T2DM. This study’s more significant variation in antibiotic resistance was associated with an increased risk of T2DM. They found that antibiotic resistance was associated with fecal metabolites. Specifically, the authors’ Diabetes-ARG score (DAS) and D-Ala-D-Ala-dipeptidase required for vancomycin resistance (Vancomycin vanX) were positively associated with Branch Chain Amino Acids (BCAA) L-isoleucine and L-leucine, which have been implicated in diabetes risk in earlier studies [43,44]. Associations were also observed between DAS and the multidrug resistance protein (Multidrug emrE). They were positively related to dihomo-gamma-linolenic acid (DGLA), which was previously found to be positively associated with obesity, body fat accumulation, and insulin resistance in patients with type 2 diabetes [45,46], while negatively associated with butyric acid, a key metabolite of healthy gut microbiota [47].

A decreased ability of the microbiome to biosynthesize SCFAs and increased production of BCAAs were observed in CMD. Fromentin et al. [34] found that these microbiome features were associated with an increased risk of asymptomatic coronary atherosclerosis. They studied individuals with ischaemic heart disease (IHD), including those with acute coronary syndrome, as well as individuals with chronic ischemic heart disease and those with IHD and heart failure. A reduction in gut bacterial cell density and a shift in the abundance of many species and in microbial functional potential were found, which seem to reverse after IHD treatment. Changes in the metabolome accompanied shifts in the microbiome. Metabolites with vasoprotective and antioxidant properties such as fatty acid esters, ergothioneine, and alpha-tocopherol were depleted, while proatherogenic trimethylamine intermediates and compounds derived from tryptophan and phenylalanine were enriched. In contrast, other metabolites, such as 4-cresol and phenylacetylglutamine, reflected the early stage of IHD.

Nonalcoholic fatty liver disease (NAFLD) is associated with lipid accumulation in the liver and is considered a risk factor for cardiovascular events [48]. Zeybel et al. [37] in a placebo-controlled 10-week clinical trial studied the safety and efficacy of combined metabolic activators (CMAs) such as L-carnitine tartrate, nicotinamide, riboside, and N-acetyl-l-cysteine concerning lipid accumulation in the liver. This small cross-omics study demonstrated the efficacy of CMA in treating NAFLD and studied the mechanisms of host-microbiota interactions. The treatment with CMAs led to a significant decrease in the abundance of fecal microbiome species belonging to the phyla *Proteobacteria*, *Actinobacteria*, and *Firmicutes* and of the oral microbiome belonging to *Proteobacteria*, *Bacteroidetes*, and *Actinobacteria*. *Faecalibacterium prausnitzii* was positively related to CMA directly related metabolites such as cysteine, cysteinyl glycine, sarcosine, and N1-methylinosine, and plasma cysteine was positively related to the abundance of *Roseburia faecis* and *Oscillibacter* sp. 57 20, which belong to the phylum *Firmicutes*, and with *Bacteroides ovatus* and *Bacteroides fragilis*, which belong to the phylum *Bacteroidetes*. Plasma metabolites indirectly associated with CMA, N1-methyl-4-pyridone-3-carboxamide, and N1-methyl-2-pyridone-5-carboxamide, were positively associated with the abundance of *Alistipes shahii* and negatively related to the abundance of *Bacteroides cellulosilyticus* and *Fusicatenibacter saccharivorans*.

Metformin, an oral antidiabetic medication, is often prescribed to treat T2DM. Earlier studies found that metformin treatment may be responsible for gastrointestinal side effects in healthy volunteers [49]. In line with these findings, Tian et al. [38] found that metformin treatment in patients with stable coronary artery disease combined with diabetes mellitus may alter gut microbiota signatures. They found that gene richness significantly increased in the microbiome of metformin-treated patients, showing an increase in unclassified *Clostridium* spp. and a reduced abundance of *Prevotella bryantii*, *Citrobacter koseri*, and *Acidaminococcus fermentans*. This study demonstrated that metformin may interact with the gut microbiota, confounding gut dysbiosis and increasing the potential for nitrogen metabolism.

In metabolically dysfunctional C57BL/6 mice, two antidiabetic medications, metformin, and the dipeptidyl peptidase-4 (DPP-4) inhibitor PKF-275-055, were studied in relation to the cecal microbiota and the markers of cardiometabolic disease [41]. Although both medications decreased *Firmicutes*/*Bacteroidetes* ratios and showed similar benefits on metabolic features such as mesenteric adiposity, microbiota, and metabolomic profiles, they differed significantly. Metformin reduced α-diversity, a metric frequently associated with host metabolic fitness, while favoring these microbiotas associated with metabolic health. It also increased plasma sphingolipids and colonic bacteria associated with sphingolipid metabolism. Earlier studies demonstrated that sphingolipid species are associated with reduced insulin sensitivity [50]. On the other hand, PKF-275-055-treated mice showed increased butyrate and acetogenic bacteria production and a reduced concentration of some sphingolipids.

### 3.2. Nutrition, Food, and Supplements

The nutrition and food articles that met the inclusion criteria spanned the years 2015–2022. Except for one publication that focused on the colonization of mice with human fecal communities, all the others were related to humans (healthy, overweight, obese, elderly, twins, and children). Publications covered the usual diet, Western diet, Mediterranean diet, calorie restriction, low-carbohydrate diet, plant-based diet, dairy intake, transition from low-fat to high-fat and low-carbohydrate diet, dietary fiber intake, raspberries, berberine, probiotics, sodium and potassium, and soy. The articles have been collected and are presented in Table 2.

#### 3.2.1. Nutrition

Obesity is a known cardiometabolic risk factor, and a poor diet can promote body fat accumulation [70]. Pallister et al. [68] conducted a cohort study of twins in which they found that visceral fat mass (VFM) and the higher dietary VFM-risk score developed by the authors were positively related to the bacterial species *Eubacterium dolichum* and plasma alpha-hydroxyisovalerate and butyrylcarnitine. According to the authors, *E. dolichum* may be a link between VFM and a diet low in fruit, whole grains, and fermented dairy products and high in red, processed meat, eggs, and fried and fast foods. In animal studies, *E. dolichum* was associated with a Western-type diet and adverse metabolic outcomes [71,72]. Alpha-hydroxyisovalerate and butyrylcarnitine are metabolites of branched-chain amino acid catabolism and fatty acid metabolism, which are associated with obesity and the risk of diabetes [73,74,75,76]. In this study, VFM was negatively associated with plasma hippurate and bilirubin. Bilirubin is an endogenous antioxidant [77] that protects against adiposity [78]. Hippurate in this study was associated with increased intakes of fruit and whole-grain products.

The latest findings by Bombin et al. [51] show that salivary bacteria of healthy lean and obese individuals may differ by phyla and composition and better reflect obesity, overweight, or lean body mass than those of fecal microbiota. The salivary bacterial composition, namely *Campylobacter, Aggregatibacter, Veillonella*, and *Prevotella*, best characterizes obese individuals. Obesity was positively associated with strong correlations, mainly between *Fretibacterium* and *Tannerella*, bacterial taxa that are involved in metabolic disorders. The dominant bacterial genera in the feces that characterized the obese group were *Agathobacter* and *Parabacteroides*. Sweeteners, primarily xylitol, have been found to affect the composition and phylogenetic diversity of the salivary and fecal microbiota. Calorie-restricted diets may produce significant changes in the microbiota and cause a shift toward more beneficial metabolite profiles [55,56]. Benítez-Páez et al. [55] studied the effects of caloric restriction with fiber supplementation in a 12-week clinical trial involving 80 overweight participants. They found that dietary interventions may depend on gender and be more beneficial for women than for men, and this effect was associated with specific changes in the intestinal microbiome and metabolome. During the study, the caloric-restricted diet reduced an abundance of microbiota, particularly *Clostridia* and *Bacteroides*, while supplementation with dietary fiber lowered blood pressure, which was associated with changes in the intestinal microbiome and metabolome. In contrast to calorie-restricted diets, low-carbohydrate diets may cause different changes in intestinal microbiota and microbiota-produced metabolites [56]. Ma et al. [56] found that, in contrast to a calorie-restricted diet, a low-carbohydrate diet can increase the *Bacteroidetes*/*Firmicutes* ratio, decrease branched-chain amino acid biosynthesis, and increase serine biosynthesis. Both calorie-restricted and low-carbohydrate diets increased acylcarnitines, which are suggested markers of cardiometabolic risk [79,80]. Conversely, a transition from a low-fat to a high-fat and low-carbohydrate diet may be associated with adverse changes in the composition of the gut microbiota, fecal metabolites, and pro-inflammatory markers in plasma [66]. The high-fat diet increased bacterial genera such as *Alistipes* and *Bacteroides* and fecal contents of arachidonic acid and lipopolysaccharides. On the other hand, a low-fat diet was associated with higher α-diversity and increased abundance of *Blautia* and *Faecalibacterium*, while reducing uremic toxins such as p-cresol and indole and increasing SCFA production. 

Accumulating evidence from murine and human studies indicates possible links between obesity and dysbiosis [25,81,82]. Zhang et al. [69] conducted a dietary intervention using diets rich in non-digestible fermentable carbohydrates in children with simple and genetic obesity who were found to be dysbiotic. They found that the gene richness and diversity of the gut microbiota decreased after the intervention period, indicating a more healthy intestinal microbiota with a greater number of carbohydrate-fermenting bacteria. Meanwhile, urine samples showed reduced levels of TMAO and indoxyl sulfate, harmful compounds resulting from lipids and proteins’ intestinal fermentation.

The Mediterranean diet (MD) is the most studied and considered one of the healthiest dietary plans for preventing and treating cardiometabolic diseases [83,84]. MD is abundant in nutrients and dietary fiber, contrary to the Western diet, which is low in nutrients and fiber but high in saturated fat, sugar, and salt. Barber et al. [53] studied fiber-rich MD (FMD) in contrast to a Western-type diet (WD) in 20 healthy men for two weeks in a randomized cross-over study. Surprisingly, fecal microbiota composition in FMD and WD groups was similar at the end of the study, suggesting little influence of diet on the habitual microbiota. FMD was associated with a higher β-diversity and a higher number of butyrate-producing bacteria compared to WD. Interestingly, despite the similarity in microbiota composition, both diets affected gut microbiota metabolism and urinary metabolite profiles differently. After the FMD, metabolic pathways studied showed higher expression compared to WD. According to the authors, these findings suggest that residential microbiota can easily adapt to dietary shifts by changing their metabolic functions. Examples of such changes caused by FMD are increased urinary excretion of TMAO after consumption of choline-rich legumes and increased urinary concentrations of metabolites of carnitine and tyramine after consumption of meat and cheese in WD, as well as metabolites with proinflammatory potential such as metabolites of cortisol and prostaglandins. Another MD study by Galie et al. [57] was carried out in overweight/obese individuals with metabolic syndrome (MetS) for two months. The research identified two clusters of microorganisms showing different metabolic patterns depending on the diet. The first one, which was associated with MD, constituted fecal bacteria belonging to the type Firmicutes and the family *Prevotellaceae*. Of the 378 plasma metabolites studied, 65, mainly lipids, acylcarnitines, amino acids, steroids, and intermediates of tricarboxylic acids, were found to be associated with MD. This metabolic profile was associated with metabolic improvement in MetS. The second cluster that seemed to reflect nuts’ consumption was found for the non-MD diet supplemented with nuts and constituted *Oxalobacter* and genera from *Christensenellaceae* and *Clostridiales*. Another 2-month study carried out on overweight/obese healthy individuals with cardiometabolic risk who shifted their dietary patterns from the Western to the Mediterranean diet showed that higher adherence to the MD pattern was associated with increased fecal concentrations of SCFAs [64]. Although microbial richness was preserved, it was found that MD dynamically modulated the gut microbiome composition and that the microbiome changes followed the increases in MD adherence.

Dietary patterns significantly impact fecal microbiota, microbiota-derived metabolites, and cardiometabolic health. Plant-based diets promote butyrate-producing microbiota. Asnicar et al. [61] examined more than 1200 microbiomes from non-diseased individuals as well as their long-term habitual dietary patterns in the large-scale Personalized Responses to Dietary Composition Trial (PREDICT 1). They divided dietary patterns into more and less healthy categories, e.g., plant-based patterns vs. less healthy plant foods and animal products. In this study, bacterial taxa such as *Roseburia hominis*, *Agathobaculum butyriciproducens, Faecalibacterium prausnitzii*, and *Anaerostipes hadrus* were associated with the consumption of healthy foods. Less-healthy diets are correlated with *Clostridia* (*Clostridium innocuum, Clostridium symbiosum, Clostridium spiroforme, Clostridium leptum,* and *Clostridium saccharolyticum*). Of the metabolites tested, circulating monounsaturated fatty acids (MUFAs), as opposed to dietary MUFAs, which are a healthy part of the diet, were associated with a less healthy diet. Firmicutes CAG:170 species showed the strongest negative association with circulating MUFAs and were negatively related to proinflammatory markers, while Clostridium bolteae showed the strongest positive association. Li et al. evaluated the relationship between a healthy plant-based diet and metabolic risk, gut microbiota, and plasma metabolites in a smaller cohort study of older men [58]. This study identified several bacterial species and metabolic pathways that were associated with lower metabolic risk. Examples of such associations include the abundance of F. prausnitzii and the degradation pathways of d-galacturonate I and 4-deoxyl-threo-hex-4-enopyranuronate; a higher abundance of *Bacteroides cellulosilyticus* and *Eubacterium eligens*; and metabolic pathways of amino acids and pyruvate fermentation. According to this study, a healthy plant-based diet reduces metabolic risk, and this effect may depend in part on the composition of the gut microbiota. The same group of researchers analyzed the relationship between red meat consumption, microbiota-mediated meat-derived trimethylamine N-oxide (TMAO), metabolic health among men, and the potential role of gut microbiota in this relationship [54]. A growing number of studies have identified TMAO and its precursors (choline and carnitine) as cardiometabolic risk factors [85,86,87,88]. Li et al. [54] identified 10 bacterial species, including *Firmicutes* species (*Clostridium citroniae, C. nexile, C. clostridioforme, Clostridiales bacterium* 1 7 47FAA, *Eubacterium hallii, E. biforme, Erysipelotrichaceae bacterium* 21–3, and *Roseburia hominis*), one *Bacteroidetes* species (*Alistipes shahii*), and one *Actinobacteria* species (*Eggerthella* unclassified) that have been linked to TMAO production. In this study, higher habitual consumption of red meat and choline increased TMAO; however, only among those participants whose microbial profiles consisted of abundant species that predicted TMAO concentrations. Of these species, *Alistipes shahii* was the most potent contributor to associations between the consumption of red meat and TMAO production.

Other animal-derived foods, such as dairy products, may presumably reduce cardiometabolic risk by affecting the gut microbiota and the metabolites it produces. Shuai et al. [60] in a prospective cohort study of 1780 middle-aged and elderly Chinese participants with a relatively low intake of dairy products (milk and yogurt) found that habitual consumption of dairy products was positively associated with gut microbiome diversity and the abundance of genera, which were positively related to lipid and metabolomic profiles showing a beneficial effect. In the group with the highest intake of yogurt, unclassified genera of the families *Ruminococcaceae, Rikenellaceae*, and *Barnesiellaceae* were enriched. In this study, a microbiota-derived metabolite, the amino acid L-alanine, was negatively associated with the yogurt-microbial score. Previous studies have found a positive association between alanine and the risk of T2DM and CVD [89,90]. Other microbiota-derived metabolites, such as 2-hydroxy-3-methylbutyric acid and 2-hydroxybutyric acid, were negatively associated with dairy consumption and dairy-microbial scores. In recent studies, 2-hydroxybutyric acid has been shown to be associated with impaired glucose metabolism [91,92].

Research suggests that dietary sodium affects the composition and function of the fecal microbiota [93]. A longitudinal study conducted by Wang et al. [63] in 392 adults with habitual excessive sodium intake and deficient potassium intake has shown that dietary sodium is associated with microbiota related to inflammation and CVD factors, such as *Dorea, Ruminococcus, Ruminococcaceae*, and *Lachnospiraceae*. In this study, microbiota-derived phenolic metabolites resulting from dietary polyphenols, such as 1,2,3-benzenetriol sulfate, 3-methoxycatechol sulfate, and 4-methylcatechol sulfate, were negatively related to sodium intake, while SCFAs such as butyrate/isobutyrate and isovalerate were positively associated with the Na/K ratio. Previous studies have shown conflicting results regarding SCFA. However, some studies have pointed to elevated fecal and plasma SCFA concentrations as positively associated with obesity [94].

#### 3.2.2. Food and Supplements

The few articles identified in this review using the search criteria focused on specific foods (raspberries, soy, dairy products) and supplement ingredients (dietary fiber, berberine, probiotics).

Food is a source of many nutrients and secondary metabolites (polyphenols, fibers, etc.) that can have beneficial effects on humans and animals. Valuable components of raspberries include vitamin C, fiber, and polyphenols (ellagitannins and anthocyanins) [95]. Raspberries exhibit beneficial properties for cardiometabolic health [96,97]. The results of the studies cannot always be interpreted unambiguously due to the heterogeneity of the study samples. Therefore, Franck et al. [52] used a transcriptomic technique to identify individuals whose raspberry consumption was associated with metabolic effects and those who did not respond to the intervention. In this 8-week intervention study of 24 responders and non-responders to raspberries, the *Firmicutes* to *Bacteroidetes* ratio was found to be significantly lower in responders at week 0 and unchanged at week 8, while it decreased in non-responders after the intervention. After consuming raspberries, the responders’ plasma lipidomic profile was characterized by a significant reduction in triglycerides and an increase in phosphatidylcholine.

One human study examined interactions between habitual soy intake, the gut microbiome, and metabolites in plasma and feces [65]. Research shows that the consumption of soy products is associated with better cardiometabolic health in humans and animals [98,99]; however, the mechanisms of this effect have not been completely investigated. Regarding this, Shah et al. [65] found that reactivity to soy may depend on the microbiome’s composition. They identified two bacterial taxa, such as *Prevotella* and *Dialister*, with different effects on systolic blood pressure. Although both taxa were suppressed by soy consumption, *Prevotella* showed a positive correlation with blood pressure, as opposed to *Dialister*. In terms of metabolites, *Prevotella* was associated with bacterial-derived tryptophane metabolites, indoleacetyl glutamine, enrichment in sphingolipid and nitrogen metabolism, and the biosynthesis of aminoacyl-tRNA.

Dietary fiber is mostly polysaccharide polymers undigested in the intestines but fermented by the gut microbiota, being a source of energy and nutrients for colonies of microbes, humans, and animals. As a recent study shows, the host’s metabolic response may depend on the type of dietary fiber and on the composition of the gut microbiome [59]. Murga-Garrido et al. [59] found that in germ-free mice colonized with two distinct human fecal communities (low-butyrate and high-butyrate-producing bacteria), fermentable fibers (cellulose, inulin, pectin, and mixed fibers) elicited different metabolic responses. The microbiota, which was engaged in most of the associations between microbiota and metabolic pathways, represented the *Firmicutes* phylum. For example, the genus *Anaerotruncus* was positively associated with some metabolites of the lipid metabolic pathway and negatively associated with metabolites of the amino acid and nucleotide pathways, while *Ruminococcus* was negatively associated with metabolites of the amino acid and lipid metabolic pathways. Of the *Bacteroidetes*, *Parabacteroides* was inversely related to metabolites of the arginine-proline and dihydroxy fatty acid pathways. *Anaerotruncus*, *Ruminococcus*, and *Parabacteroides* showed higher abundance in mice receiving inulin colonized with microbiota producing little butyrate and were associated with adverse outcomes such as higher adiposity, liver triglycerides, and glucose compared to mice colonized with high-butyrate-producing bacteria [59].

Berberine is a naturally occurring compound that belongs to the isoquinoline alkaloids. It has been used for centuries in Chinese medicine to treat several diseases, including diabetes. The hypoglycemic effect of berberine is well known, and berberine is now found as one of the more potent ingredients in supplements. The effect of berberine (BBR) on metagenomics and metabolomics was studied by Zhang et al. [62] in a 12-week randomized clinical trial conducted in China. This study divided 409 newly diagnosed T2DM individuals into four groups: BBR-alone, probiotics + BBR, probiotics-alone, or placebo. A dual-omics approach identified a plausible mechanism underlying berberine’s hypoglycemic effect, with *Ruminococcus bromii* as a key microbial player in deoxycholic acid production. Bile acids have been known for decades as signaling molecules that activate several nuclear receptors and thus may participate in lipid metabolic pathways [100].

Probiotics support intestinal function and integrity and reduce inflammation and intestinal permeability [101]. Probiotics have recently been found to reduce body fat accumulation [102,103]; however, the mechanisms are not fully understood. Hibberd et al. [67] in a randomized, double-blind, placebo-controlled clinical trial studied the effects of the probiotic bacterial strain *Bifidobacterium animalis* subsp. Lactis 420 TM (B420) with or without the prebiotic polydextrose on fecal microbiota and fecal and plasma metabolites in overweight and obese adults. After 6 months of intervention, they found that *Lactobacillus* and *Akkermansia* were more prevalent with B420 intake, and Bifidobacterium was positively correlated with lean body mass. *Christensenellaceae* spp., correlated negatively with fat mass in the waist area and energy intake at baseline and increased after the intervention in both prebiotic and prebiotic and probiotic supplemented subjects. Variability was found in fecal metabolomics results; however, there was some reduction in plasma bile acids: glycolic acid, glycocholic acid, taurohydrocholic acid, and tauroursodeoxycholic acid, in participants supplemented with probiotics and prebiotics taken together. 

## 4. Conclusions

The literature search performed for this review indicates that the combined approach of metagenomics and metabolomics in cardiometabolic diseases is a new and expanding field. The number of papers combining metagenomic and metabolomic technologies is still limited from the perspective of cardiometabolic diseases; however, in recent years there has been a surge of studies using the combination of metagenomics and metabolomics to investigate pharmacotherapy and nutrition in cardiometabolic diseases. Most of the papers identified involved human studies, covering the last three years. Only three studies involved rodent research. While animal models may have advantages in studying host-microbiota interactions, they often do not provide reliable preclinical results that can be easily translated into effective human treatments.

There is growing evidence that drugs and foods, through metabolites of the microbiota, can modulate the cardiometabolic health of the host. A whole range of interactions occur between drugs and food, and it is now a matter of time before metagenomic and metabolomic studies examining such interactions begin.

## Figures and Tables

**Figure 1 biomolecules-13-01340-f001:**
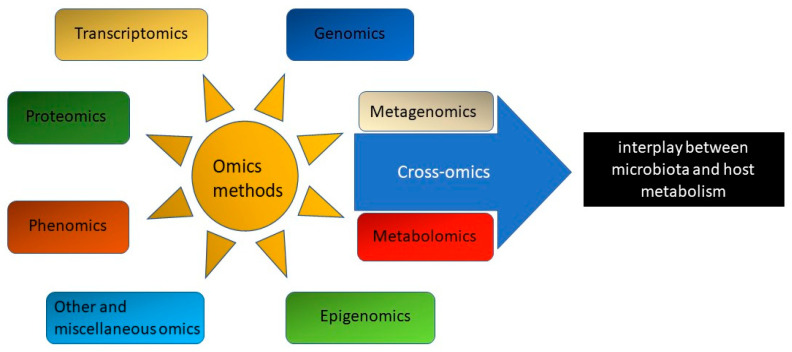
Overview of omics methods, focusing on metagenomics and metabolomics.

**Figure 2 biomolecules-13-01340-f002:**
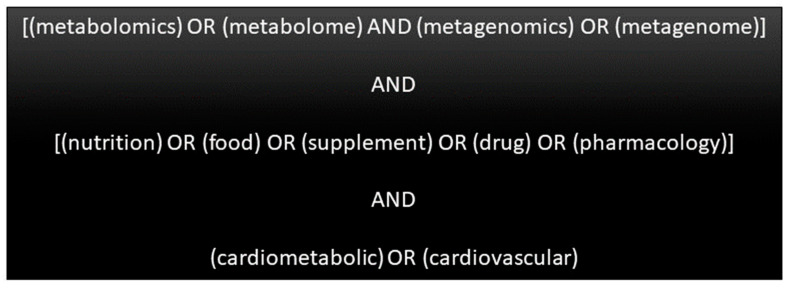
Search terms used in this review.

**Figure 3 biomolecules-13-01340-f003:**
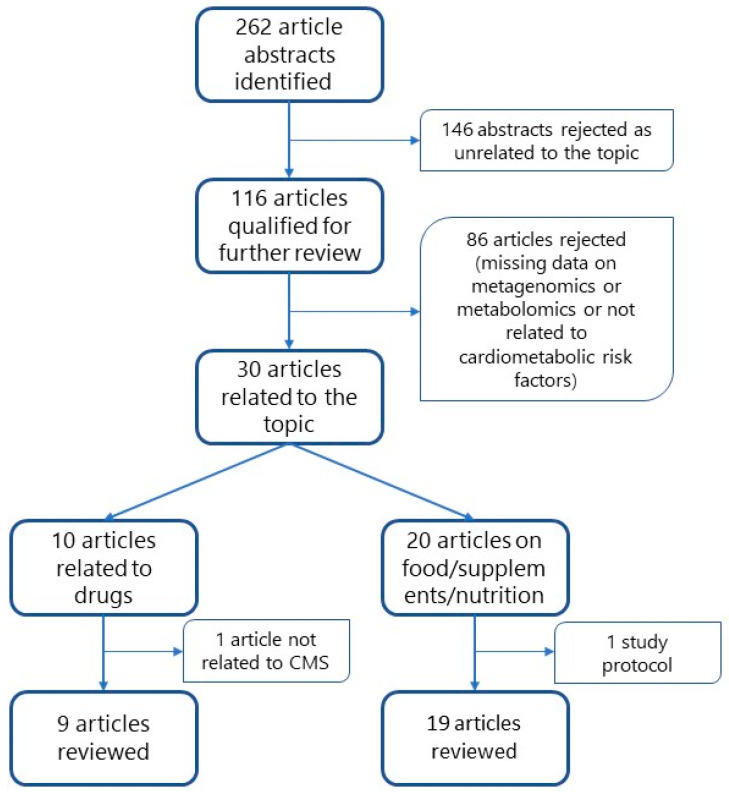
Flowchart of literature selection.

**Table 1 biomolecules-13-01340-t001:** Summary of studies ranked in order of most recent using metagenomics and metabolomics to study pharmacotherapy in CMD.

	Reference, Year	Drugs Studied	Study Design	Participants/Animals	Samples/Omics Methods	Major Findings	No Ref.
1.	Wilmanski et al., 2022	Statins	Cross-sectional cohort study	two independent cross-sectional cohorts: discovery *n* = 1848 (244 statin users); validation *n* = 991 (688 with various stages of cardiometabolic disease)	- fecal metagenome; - blood metabolomics	- HMG (a hydrolyzed substrate for 3-hydroxy-3-methylglutarate-coenzyme-A (HMG-CoA) reductase) appears as a marker for statin on-target effects - heterogeneity in statin responses is associated with variation in the gut microbiome - more intense on-target and adverse effects of statins found in microbiomes enriched with *Bacteroides* and lacking diversity.	[33]
2.	Fromentin et al., 2022	Polypharmacy	Observational study	1241 middle-aged Europeans, healthy and with ischemic heart disease (IHD)	- fecal metagenome; - serum and urine metabolome	- about 75% of microbiome and metabolome features in individuals with dysmetabolism can distinguish individuals with IHD from healthy individuals after adjustment for confounders associated with medication and lifestyle - changes in the gut microbiome and metabolome can begin long before the clinical onset of IHD - statins are associated with the restoration of diversity in the gut microbiota.	[34]
3.	Shuai et al., 2022	Antibiotics	Cohort study	1210 healthy, prediabetic, and T2D participants	- fecal metagenome; - fecal metabolome	- a shift of antibiotic resistance genes (ARGs) in groups of healthy subjects, prediabetes, and T2DM - larger ARG diversity associated with a higher risk of T2DM - study-developed ARG score associated with T2DM progression - gut ARG associated with cardiometabolic risk factors.	[35]
4.	Forslund et al., 2021	28 drugs and several drug combinations	Meta-Cardis cohort	2173 European residents	- fecal microbiome; - serum metabolome	- additive effects of drugs cause a shift of the metabolome and microbiome towards a healthier state - strongest effects on the serum metabolome are for antidiabetic drugs, statins, beta-blockers, antithrombotic drugs, and aspirin - antibiotics exhibit a quantitative relationship between the number of courses prescribed and progression towards a microbiome state associated with the severity of cardiometabolic disease - the relationship between CMD medication dosage, improvement in clinical markers, and microbiome composition, supports direct drug effects.	[36]
5.	Zeybel et al., 2021	Combined metabolic activators (CMA) (3.73 g L-carnitine tartrate, 1 g nicotinamide riboside, 12.35 g serine, and 2.55 g N-acetyl-l-cysteine)	10-week placebo-controlled phase 2 study to investigate the efficacy and safety of CMA	NAFLD patients: 20 in the CMA treatment group and 11 in the placebo group	- oral and fecal metagenome; - plasma metabolome	- a significant decrease in the abundance of species belonging to *Proteobacteria*, *Actinobacteria*, and *Firmicutes* in the fecal microbiome after the CMA treatment - the abundance of specific species of *Proteobacteria, Bacteroidetes*, and *Actinobacteria* was reduced in the oral microbiota in the CMA group - N-trimethyl-5-aminovalerate associated with intestinal microbiota was the most significantly reduced metabolite in the CMA group and was significantly lower than in the placebo group - CMA treatment positively correlated with the plasma levels of serine, glycine, gamma-glutamylglycine, carnitine, 1-methylnicotinamide, N1-methyl-4-pyridone-3-carboxamide, and N1-methyl-2-pyridone-5-carboxamide.	[37]
6.	Tian et al., 2021	metformin	Prospective cohort	- 71 patients with stable coronary artery disease (SCAD) - 38 SCAD + T2DM - 55 healthy control	fecal and serum samples	- a significant difference in gut bacteria between SCAD and SCAD + T2DM patients - metformin may confound gut dysbiosis and increase the potential for nitrogen metabolism.	[38]
7.	Jin et al., 2021	Orlistat, ezetemibe	Randomized controlled open-label trial	overweight and obese individuals with dyslipidemia: - 37 taking orlistat - 31 taking ezetimibe	- fecal metagenome; - fecal metabolome	- intestinal malabsorption of dietary fat and cholesterol caused by orlistat and ezetimibe had limited effect on the overall gut microbial community and their metabolites.	[39]
8.	Kappel et al., 2020	oral antibiotics	Animal study, human study	- ApoE-knockout mice - 42 humans with carotid atherosclerosis	- serVum metabolome; - cecVal microbiome	- antibiotic therapy in mice had an adverse effect on the development of atherosclerosis regardless of diet - humans with atherosclerosis showed a trend toward lower α-diversity, lower levels of tryptophan, and higher levels of long-chain fatty acids.	[40]
9.	Ryan et al., 2020	metformin and dipeptidyl peptidase-4 (DPP-4) inhibitor, PKF-275-055	Animal study	C57BL/6 male mice	- cecal microbiota; - plasma metabolome	- microbiota and metabolomic profiles differed between metformin and PKF-275-055-treated mice - metformin and PKF-275-055 treatment decreased Firmicutes/Bacteroidetes ratios - metformin favors metabolic health-associated *Akkermansia, Parabacteroides,* and *Christensenella* - metformin reduced α-diversity, a metric frequently associated with host metabolic fitness–PKF-275-055 treatment increased levels of butyrate-producing *Ruminococcus* and acetogen *Dorea*, with reduced levels of certain plasma sphingomyelin, phosphatidylcholine, and lysophosphatidylcholine entities. In turn, metformin reduced levels of acylcarnitines, a functional group associated with systemic metabolic dysfunction. Finally, several associations were identified between metabolites and altered taxa.	[41]

**Table 2 biomolecules-13-01340-t002:** Summary of studies ranked in order of most recent using metagenomics and metabolomics to investigate nutrition and consumption of specific foods in CMD.

	Reference, Year	Diet/Food Studied	Study Design	Participants/Animals	Samples/Omics Methods	Major Findings	No Ref.
1.	Bombin et al., 2022	Habitual diet	Observational study	135 healthy individuals (lean, overweight, obese)	- fecal and salivary metagenomics; - plasma and fecal metabolomics	- salivary bacterial communities differ by composition and phyla in lean and obese individuals - An increase in obesity status is positively associated with strong correlations between bacterial taxa, mainly with bacterial groups implicated in metabolic disorders, including *Fretibacterium* and *Tannerella* - Consumption of sweeteners, especially xylitol, significantly influences the compositional and phylogenetic diversities of salivary and fecal microbiota.	[51]
2.	Franck et al., 2022	Raspberry consumption	8-week randomized controlled trial	24 participants (responders and non-responders to raspberries according to transcriptional profiles)	- fecal metagenomics; - serum metabolomics	- distinct metagenomic profile identified before intervention in responders and non-responders - lower *Firmicutes*-to-*Bacteroidetes* ratio found in responders compared to non-responders - plasma lipidomic profile of responders was characterized by a significant decrease in triglycerides and an increase in phosphatidylcholines following raspberry consumption.	[52]
3.	Barber et al., 2022	Western-type diet (WD) vs. a fiber-enriched Mediterranean diet (MD)	2-week randomized cross-over study	20 healthy men	- fecal metagenomics; - urinary metabolomics	- relatively little difference in microbiota composition between MD and WD - microbial metabolism differed substantially, as shown by urinary metabolite profiles and the abundance of microbial metabolic pathways - effects of the diet were less evident in individuals with higher β diversity.	[53]
4.	Li et al., 2022	Habitual diet	Longitudinal cohort study	307 healthy men, Men’s Lifestyle Validation Study	- fecal metagenomics; - urine and plasma metabolomics	- 10 microbial species significantly associated with plasma trimethylamine N-oxide (TMAO) - higher habitual intake of red meat and choline significantly associated with increased TMAO among men with a microbial profile of abundant species predicted TMAO concentrations. - -*Alistipes shahii* species significantly strengthened the positive associations of red meat intake with TMAO concentrations and HbA1c.	[54]
5.	Benítez-Páez et al., 2021	Caloric restriction with fiber supplementation	12-week clinical trial	80 overweight participants	- fecal metagenomics; - plasma and fecal lipidomics	- an abundance of microbial species, mainly *Clostridia* and *Bacteroides*, reduced by the caloric-restricted diet - supplementation of caloric-restricted diets with soluble fiber lowered blood pressure in all participants and had a more significant effect on women than on men, which was associated with specific changes in the gut microbiome and metabolome - fecal lithocholic acid was significantly reduced in women, and this effect was likely to be related to the presence of *Clostridia comes* or *Roseburia torques.*	[55]
6.	Ma et al., 2021	Low-carbohydrate (LC) and calorie-restricted (CR) diets	12-week clinical trial	48 overweight and obese women	- fecal metagenomics; - erythrocyte and plasma metabolomics	- LC and CR diets produced different changes in the gut microbiota, plasma acylcarnitines, and erythrocyte fatty acids - the *Bacteroidetes* to *Firmicutes* ratio increased significantly in the LC diet but not in the CR diet - LC group vs. CR group had lower BCAA biosynthesis and higher serine biosynthesis.	[56]
7.	Galié et al., 2021	Mediterranean diet (MD) or a non-MD diet supplemented with 50 g nuts	2 months randomized controlled cross-over trial	44 adults with overweight/obesity and metabolic syndrome	- fecal metagenomics; - plasma metabolomics	- -MedDiet produced significant changes in 65 circulating metabolites, mainly lipids, acylcarnitines, amino acids, steroids, and intermediates of the tricarboxylic acid - -two clusters of microbial genera with opposing behaviors toward selected metabolites were identified, mainly species of phosphocholine, cholesteryl esters, triglycerides, and medium- and long-chain acylcarnitines.	[57]
8.	Li et al., 2021	plant-based diet index (hPDI)	Longitudinal cohort study	303 older men, Men’s Lifestyle Validation Study	- fecal metagenomics; - plasma metabolomics	- -the hPDI was associated with the relative abundance of seven species and nine pathways - -higher hPDI was associated with a higher relative abundance of *Bacteroides cellulosilyticus* and *Eubacterium eligens*, amino acid biosynthesis pathways (l-isoleucine biosynthesis I and III and l-valine biosynthesis), and the pyruvate to isobutanol fermentation pathway.	[58]
9.	Murga-Garrido et al., 2021	Dietary fibers: cellulose, inulin, pectin, a mix of 5 fermentable fibers	Animal intervention study	66 genetically-identical germ-free mice colonized with two distinct human fecal communities	- fecal metagenomics; - serum metabolomics	- -ingesting the same type of dietary fiber by genetically identical mice but differing in microbiomes transferred from humans can lead to various metabolic outcomes in the host - -fiber-specific differences were found in the intestinal levels of butyrate and valerate - -genus *Anaerotruncus* was negatively associated with long-chain saturated, unsaturated, and branched fatty acids, lysophospholipids, and monoacylglycerol, and positively with lysine, glycine, arginineproline, purine, and pyrimidine - -*Ruminococcus* was negatively associated with BCAA, glutamate, tryptophan, fatty acids, purine, and gamma-glutamyl amino acid.	[59]
10.	Shuai et al., 2021	Habitual dairy consumption	Prospective cohort study	1780 participants in Guangzhou Nutrition and Health Study	- fecal metagenomics; - serum metabolomics	- an overall difference in gut microbial community structure (β-diversity) between the highest and lowest categories of dairy, milk, and yogurt - association of targeted serum metabolites with dairy-microbial features and cardiometabolic traits - 2-hydroxy-3-methylbutyric acid, 2-hydroxybutyric acid, and L-alanine were inversely associated with the dairy-microbial score.	[60]
11.	Asnicar et al., 2021	Habitual diet	Clinical trial	1203 gut microbiomes from 1098 individuals of Personalised Responses to Dietary Composition Trial (PREDICT 1)	- fecal metagenomic; - blood metabolomics	- taxa associated with healthy plant-based foods included mainly butyrate - producers (*Roseburia hominis, Agathobaculum butyriciproducens, Faecalibacterium prausnitzii, and Anaerostipes hadrus*) - less healthy plant and animal products, which were correlated with *Clostridia* (*Clostridium innocuum, Clostridium symbiosum, Clostridium spiroforme, Clostridium leptum, and Clostridium saccharolyticum*).	[61]
12.	Zhang et al., 2020	Berberine, probiotics	12-week randomized double-blind clinical trial	409 T2DM individuals	- fecal metagenomic; - plasma metabolomics	- hypoglycaemic effect of berberine in T2DM was mediated by the inhibition of deoxycholic acid biotransformation by *Ruminococcus bromii*	[62]
13.	Wang et al., 2020	Sodium, potassium, and Na/K ratio	Longitudinal study, China Health and Nutrition Survey	392 adults	- fecal metagenomics; - plasma metabolomics	- dietary sodium, potassium, and Na/K ratios were associated with microbiota and metabolites related to inflammation and CVD risk factors - dietary sodium was inversely associated with the phenolics 1,2,3-benzenetriol sulfate, 3-methoxycatechol sulfate, and 4-methylcatechol sulfate derived from the microbial conversion of nutritional polyphenols and with SCFAs (butyrate/isobutyrate and isovalerate) - *Coriobacteriaceae* and *Ruminococcaceae* were positively associated with 4-methylcatechol sulfate - *Coriobacteriaceae* was involved in the phenolic conversion.	[63]
14.	Meslier et al., 2020	Mediterranean diet (MD)	8-week randomized controlled trial	82 healthy overweight and obese subjects with a habitually low intake of fruit and vegetables and a sedentary lifestyle	- metagenomics - fecal, urinal, and serum metabolomics.	- MD resulted in increased levels of the fiber-degrading bacteria Faecalibacterium prausnitzii and of genes for microbial carbohydrate degradation associated with butyrate metabolism - shift to MD resulted in increased urinary urolithins, fecal bile acid degradation, and insulin sensitivity, which co-occurred with specific taxa of microorganisms.	[64]
15.	Shah et al., 2020	Soy intake	Cross-sectional study	104 healthy lean, overweight, or obese subjects	- fecal metagenomics, - plasma and fecal metabolomics	- reduction in blood pressure in response to soy may be dependent on microbiome composition - -suppression of genus *Dialister* and *Prevotella* by high soy consumption.	[65]
16.	Wan et al., 2019	The transition from a low-fat diet to a high-fat and low-carbohydrate diet	6-month randomized controlled trial	217 healthy young adults	- fecal metagenomics; - fecal metabolomics	- the lower-fat diet was associated with increased α-diversity, increased abundance of *Blautia* and *Faecalibacterium,* and decreased p-cresol and indole - the higher-fat diet was related to increased *Alistipes, Bacteroides,* arachidonic acid, and the lipopolysaccharide biosynthesis pathway and decreased *Faecalibacterium* and total SCFA.	[66]
17.	Hibberd et al., 2019	Probiotic (*Bifidobacterium animalis* subsp. lactis 420™), prebiotic (polydextrose)	Randomized, double-blind, placebo-controlled clinical trial	134 overweight and obese adults	- fecal metagenomics; - fecal and plasma metabolomics	- supplementation with probiotics and prebiotics increased *Akkermansia, Christensenellaceae,* and *Methanobrevibacter* and decreased *Paraprevotella* - plasma bile acids glycocholic acid, glycoursodeoxycholic acid, taurohyodeoxycholic acid, and tauroursodeoxycholic acid were reduced in probiotic- and prebiotic-supplemented individuals.	[67]
18.	Pallister et al., 2017	Habitual diet	Cohort study	2218 twins	- blood metabolomics; - fecal metagenomics	- visceral fat mass (VFM) score and VFM were associated with *Eubacterium dolichum* and four blood metabolites (hippurate, alpha-hydroxyisovalerate, bilirubin, and butyrylcarnitine.	[68]
19.	Zhang et al., 2015	A diet rich in non-digestible carbohydrates	30-day dietary intervention for body weight reduction	17 children with Prader-Willi syndrome (PWS), 21 children with simple obesity	- fecal metagenomics; - urine and fecal metabolomics	- enriched pathways for carbohydrate metabolism and decreased pathways for fat and protein metabolism after the intervention - 13 bacterial Co-Abundance Gene Groups (mostly in *Ruminococcus* spp., *Parabacteroides spp.* and *Bacteroides* spp.) had gene clusters for anaerobic choline degradation.	[69]

## Data Availability

Not applicable.

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
