# Peer review of "Pharmacological and Nutritional Modulation of Metabolome and Metagenome in Cardiometabolic Disorders"

_biomolecules, 2023, doi:10.3390/biom13091340_

Round 1

Reviewer 1 Report

This article seems to be an exhaustive review of the subject treated.
However: 1) the few animal studies do not seem to me to have their place here,
2) a summary table would be very useful because there is a large mass of data here;
in particular, a clear distinction between primary and secondary prevention (or between patients at low or high cardiovascular risk)
would make the messages more accessible and/or usable for the readers

Author Response

We appreciate your time spent reviewing our manuscript and for your pertinent comments. We greatly appreciate it.

We have tried to address and discuss your comments as thoroughly as possible, however:

1) the few animal studies do not seem to me to have their place here,

We understand the reviewer's position and have repeatedly considered including animal studies in this review before writing the manuscript. Finally, we decided that the review aimed to find out what studies were conducted from the beginning on combined metabolomics and metagenomics methods in the context of CMD risk factors, regardless of whether they were conducted on humans or animals.

The removal of animal studies would disrupt the entire workflow, as it would require the use of the additional research term 'human' and the repetition of studies with the added term, as the whole number of studies would change, as well as the flow pattern thereafter. The research works found indicate that these are mainly human studies. Therefore, this would hint to researchers conducting animal studies that this is an under-exploited topic. Furthermore, some studies described in this review combine both animal and human studies (Kappel 2020, Murga-Garrido 2021). It would therefore be questionable to remove these studies from the review. As animal studies are few in this review but contribute important quality, we would ask for your understanding that we have left them in this review.

2) a summary table would be very useful because there is a large mass of data here; in particular, a clear distinction between primary and secondary prevention (or between patients at low or high cardiovascular risk) would make the messages more accessible and/or usable for the readers

The reviewer's comment is factual and we ourselves reflected on the issues raised by the reviewer before preparing the manuscript for submission. We understand that there is a lot of material in the manuscript and an additional division would make the work easier to read. Therefore, we tried to meet the reviewer's expectations and started to prepare a summary table, but our attempts to make a division were not successful, as the studies included in the review contain very different research patterns, e.g. there are healthy subjects and patients in the studies at the same time; sick people with different stages of the disease, e.g. pre-diabetes, diabetes; some studies are observational and others are intervention studies. Therefore, the most sensible option seemed to us not to add a summary table.

Reviewer 2 Report

The work written by Witkowska and Salem summarizes the literature research on metabolomics and metagenomics results in cardiovascular diseases. 

I personally think the review is scientifically loud and is generally well written.

Please find some comments below:

References

In general, while reading the text, references were mentioned in a later statement. I suggest the authors to check that. Examples are in lines 46, 47, 58, 63. In addition, in line 138 please add the reference to the one publication mentioned in the text.

Introduction

a.     I suggest the authors to simplify/rephrase the text and better explain what CMD is.

b.     Additional information, such as incidence, prevalence and/or numbers related to its morbidities can help the reader to understand its actual impact on the population. 

c.     What is the impact that such a complex and multifactorial disease has on the health system?

Line numbers

For easier review process please reset the line numbers of the whole manuscript as from page 21, the line numbers start from 1 again. From here line 199 please add reference

Name of the sections/text

a.     I suggest renaming the Section “Nutrition and Food” as “Nutrition, Food and Supplements”. Then to call the additional sections as “Nutrition” and “Food and Supplements” 

b.     Text: please change the “more” important findings to the “most” important findings (line 132)

Table 1

Ref 1 – please include additional data on the study participants, such as if healthy/diseased/others

Ref 9 – please include additional information on the mice included in the study (did these mice have any type of genetic variant?)

Quality of English is good. I suggest minor changes.

Author Response

We appreciate your time spent reviewing our manuscript and for your pertinent comments. We greatly appreciate it.

1) I suggest the authors to simplify/rephrase the text and better explain what CMD is. Additional information, such as incidence, prevalence and/or numbers related to its morbidities can help the reader to understand its actual impact on the population.  What is the impact that such a complex and multifactorial disease has on the health system?

Thank you for your valid comments. All these comments have been used to improve this part of the manuscript.

2) Line numbers. For easier review process please reset the line numbers of the whole manuscript as from page 21, the line numbers start from 1 again. From here line 199 please add reference.

The numbering errors probably arose after the insertion of the tables by the editorial office of MDPI. We are unable to change this numbering.

A literature reference has been added to line 199.

3) Name of the sections/text

  1. I suggest renaming the Section “Nutrition and Food” as “Nutrition, Food and Supplements”. Then to call the additional sections as “Nutrition” and “Food and Supplements” 
  2. Text: please change the “more” important findings to the “most” important findings (line 132)

Table 1

Ref 1 – please include additional data on the study participants, such as if healthy/diseased/others

Ref 9 – please include additional information on the mice included in the study (did these mice have any type of genetic variant?)

All of the above reviewer comments were used to improve the manuscript.